# Beta-Glucan from *S. cerevisiae* Protected AOM-Induced Colon Cancer in cGAS-Deficient Mice Partly through Dectin-1-Manipulated Macrophage Cell Energy

**DOI:** 10.3390/ijms231810951

**Published:** 2022-09-19

**Authors:** Sulaiman Binmama, Cong Phi Dang, Peerapat Visitchanakun, Pratsanee Hiengrach, Naraporn Somboonna, Thanya Cheibchalard, Prapaporn Pisitkun, Ariya Chindamporn, Asada Leelahavanichkul

**Affiliations:** 1Center of Excellence on Translational Research in Inflammation and Immunology (CETRII), Department of Microbiology, Chulalongkorn University, Bangkok 10330, Thailand; 2Department of Microbiology, Faculty of Medicine, Chulalongkorn University, Bangkok 10330, Thailand; 3Microbiome Research Unit for Probiotics in Food and Cosmetics, Chulalongkorn University, Bangkok 10330, Thailand; 4Program in Biotechnology, Faculty of Science, Chulalongkorn University, Bangkok 10330, Thailand; 5Division of Allergy, Immunology, and Rheumatology, Department of Medicine, Faculty of Medicine, Ramathibodi Hospital, Mahidol University, Bangkok 10330, Thailand; 6Nephrology Unit, Department of Medicine, Faculty of Medicine, Chulalongkorn University, Bangkok 10330, Thailand

**Keywords:** colon cancer, *S. cerevisiae*, whole glucan particle, cGAS deficiency, macrophages, dysbiosis, Dectin-1

## Abstract

Although the impacts of *Saccharomyces cerevisiae* on cancers are mentioned, data on its use in mice with cyclic GMP-AMP synthase deficiency (cGAS-/-) are even rarer. Here, 12 weeks of oral administration of *S. cerevisiae* protected cGAS-/- mice from azoxymethane (AOM)-induced colon cancers, partly through dysbiosis attenuation (fecal microbiome analysis). In parallel, a daily intralesional injection of a whole glucan particle (WGP; the beta-glucan extracted from *S. cerevisiae*) attenuated the growth of subcutaneous tumor using MC38 (murine colon cancer cell line) in cGAS-/- mice. Interestingly, the incubation of fluorescent-stained MC38 with several subtypes of macrophages, including M1 (using Lipopolysaccharide; LPS), M2 (IL-4), and tumor-associated macrophages (TAM; using MC38 supernatant activation), could not further reduce the tumor burdens (fluorescent intensity) compared with M0 (control culture media). However, WGP enhanced tumoricidal activities (fluorescent intensity), the genes of M1 pro-inflammatory macrophage polarization (*IL-1β* and *iNOS*), and Dectin-1 expression and increased cell energy status (extracellular flux analysis) in M0, M2, and TAM. In M1, WGP could not increase tumoricidal activities, Dectin-1, and glycolysis activity, despite the upregulated *IL-1β*. In conclusion, *S. cerevisiae* inhibited the growth of colon cancers through dysbiosis attenuation and macrophage energy activation, partly through Dectin-1 stimulation. Our data support the use of *S. cerevisiae* for colon cancer protection.

## 1. Introduction

Colorectal cancer is one of the leading causes of cancer-related death worldwide [1,2]. The pathogenesis of gastrointestinal cancers consists of multifactorial factors, including some genetic background and ethnicity and environmental lifestyle factors (a high fat with low fiber diet, alcoholic consumption, smoking, and an overweight and sedentary lifestyle) [2]. Among these risk factors, the avoidance of dietary cancer promotors (several hydrocarbon and dioxin-liked compounds) [3] and the administration of some products with health benefits are most often mentioned. As such, probiotics are non-pathogenic organisms with possible health benefits, including colon cancer prevention, through several possible mechanisms such as the maintenance of the physicochemical conditions of enterocytes, the attenuation of dysbiosis (an imbalance in gut microbiota), a reduction in carcinogenic compound-producing bacteria (e.g., *Escherichia coli* and *Clostridium perfringens*) and carcinogen degradation, the enhanced production of some beneficial molecules (e.g., short-chain fatty acids), and immune modulation [4].

Indeed, *Saccharomyces cerevisiae* (*S. cerevisiae*), the regular yeasts used for several human foods, are Eukaryotic cells with several biological proteins that resemble human cells [5] and inhibit cancer growth through anti-proliferation and apoptosis induction properties [6]. Additionally, *S. cerevisiae* also induces some immune responses against cancers possibly through lymphocyte activities and trained immunity (enhanced innate immune activities) [7,8]. In parallel, the responses against cancer cells by innate immunity, especially macrophages, are important, as the tumor micro-environment induces macrophages that promote tumor growth, referred to as “tumor-associated macrophages (TAM)”, and the anti-tumor effect of *S. cerevisiae* through macrophage manipulation is mentioned [9]. Indeed, whole glucan particles (WGPs), the soluble (1→3)/(1→6)-β-glucan that is the major cell wall component of *S. cerevisiae*, enhances macrophage activity, partly through energy promotion [10], and the antitumor effect of WGP in an animal model through dendritic cell functions was reported [11]. Hence, Dectin-1, a pattern recognition receptor for glucan on the surface of myeloid cells (neutrophils, macrophages, and dendritic cells) and B cells [12], might be responsible for the tumoricidal effect through immune cell activations. Nevertheless, studies on immune modulation of *S. cerevisiae* in macrophages are few in number.

The colon cancer model with the use of azoxymethane (AOM), a metabolite of dimethylhydrazine (DMH), which is a strong DNA alkylating agent found in cycads [13], together with enterocyte damage by dextran sulfate solution (DSS) for 6 months, supports hydrocarbon ingestion with chronic mucosal inflammation being a cause of cancer [14]. Although *S. cerevisiae* attenuated DMH-induced colon cancer [15], the mechanism of action is still inconclusive. On the other hand, cyclic GMP-AMP synthase (cGAS) is the central cytosolic double-stranded DNA (dsDNA) sensor that recognized the self-DNA from the dying cells (cancer cells), allowing the innate immune system to respond against the abnormal cells [16]. Accumulated self-dsDNA in cancer cells induce a cGAS conformational change to catalyze the formation of 2’,3’-cyclic GMP-AMP (cGAMP), a cyclic dinucleotide (CDN) from ATP and GTP that activates the stimulator of interferon genes (STING), results in the local production of type-I interferon (IFN-I)-induced cell apoptosis [17]. Unsurprisingly, cGAS deficient (cGAS-/-) mice have been used as susceptible cancer models.

Because (1) *S. cerevisiae* or WGP might attenuate colon cancer through a direct effect on immune cells [10] or tumor cells [1,11,18], (2) the presence of Dectin-1 (a major receptor of WGP) in macrophages and its importance on cancer cells are well-known [19], and (3) there remains a lack of data on AOM-induced cancer in cGAS-/- mice, we hypothesized that the beta-glucan of *S. cerevisiae* (WGP) attenuated cancer, even with the more cancer-susceptible property of cGAS-/- mice, through the induction of tumoricidal macrophages and tested the hypothesis in vivo and in vitro.

## 2. Results

### 2.1. Azoxymethane Induced Colon Cancer Only in cGAS-Deficient (cGAS-/-) but Not in the Wild-Type nor S. cerevisiae-Administered cGAS-/- Mice Partly through Dysbiosis Attenuation

Spontaneous colon cancer without an alteration in body weight was demonstrated only in cGAS-/-, but not in wildtype (WT), mice, with a variation in the number of malignant lesions and the total tumor burdens (Figure 1A–C). As such, five out of nine cGAS-/- mice developed intestinal lesions, while no mice in other groups demonstrated colon cancers (Figure 1C,D), supporting the vulnerability of cGAS-/- mice against cancer development [20]. The representative gross pictures and histological characteristics (homogeneous bizarre cell morphologies) of spontaneous colon carcinoma are demonstrated in Figure 1E. Because we hypothesized that the anti-tumor effect of *S. cerevisiae* might be due to (1) dysbiosis attenuation [21] or (2) immune modification, especially an enhancement in macrophage activities [22], further experiments were conducted. Although the fecal microbiome analyses among WT versus cGAS-/- mice with or without *S. cerevisiae* administration were not obviously different, as indicated by the abundance of bacteria in phylum and species level (Figure 2A–C), the possible unique bacteria in each group using linear discriminant analysis (LDA score) (Figure 2D) were different. While *Anaeroplasma* was a feature that characterized WT feces, *Chlamydia, Sutterella, Beta-proteobacteria*, and *Mycoplasma* characterized the feces of WT with *S. cerevisiae* (Figure 2D). Likewise, *Turicibacter, Coprococcus, Peptostreptococcus, Rikenella, Clostridium, Gemella, Parabacteroides,* and *Marvinbryantia* were a feature of cGAS-/- mice, while some beneficial bacteria (*Lactobacillus* and *Akkermansia*), *Bacteroides, Verrucomicrobia,* and *Ruminococcus* characterized the feces of cGAS-/- mice with *S. cerevisiae* (Figure 2D). Additionally, nonmetric multidimensional scaling (NMDS), a statistical tool that groups data points into classes of similar points and enables complex multivariate data sets to be visualized in a reduced number of dimensions, demonstrated a possible similarity between cGAS-/- feces and cGAS-/- with *S. cerevisiae* feces (Figure 2E). In parallel, the principal coordinate analysis (PCoA) of the community structure using ThetaYC distances, a statistical method which converts data between groups into a visualization of the similarity among groups, demonstrated a possible similarity between WT feces, the feces of WT with *S. cerevisiae*, and cGAS-/- feces (Figure 2F). Although there was only a subtle difference among these groups, cGAS-/- feces was the only group that demonstrated an increase in *Rikenella* and *Turicibacter* (Figure 2G), which might be associated with colon cancer. However, the total abundance of fecal bacteria as indicated by total operational taxonomic units (OTUs) and alpha-diversity analysis (Chao-1 and Shannon score) among all groups was not different (Figure 2G). Thus, the attenuation of colon cancer by *S. cerevisiae* in the AOM-induced colon cancer model might partly be due to the impact of *S. cerevisiae* on fecal microbiome alteration.

### 2.2. Intralesional Injection of the Extract from S. cerevisiae or Whole Glucan Particle (WGP) Attenuated Tumor Growth Partly through Macropahge Responses

Because we hypothesized that beta-glucan on the *S. cerevisiae* cell wall (WGP) attenuates cancers in cGAS-/- mice, the subcutaneous injection of cancer cell line (MC38) with the intralesional injection of *S. cerevisiae* crude extract (yeast extract), WGP, or normal saline solution (NSS) control was performed in cGAS-/- mice (Figure 3A). Indeed, both yeast extract and WGP attenuated tumor growth at the 4^th^ week of the experiments (Figure 3B), with the reduction in serum IL-1β and IL-6, but not TNF-α and IL-10 (Figure 3C–F), supporting a previous publication [1]. Despite the deficiency in cGAS, *S. cerevisiae* still attenuated cancers, implying a cGAS-independent mechanism that might be correlated with WGP-altered macrophage activities. Because WGP might have a different influence on different subtypes of macrophages, WGP was incubated in several macrophage manipulation protocols, including M0 (control), M1 (activation by lipopolysaccharide; LPS), M2 (IL-4 stimulation), and TAM (tumor supernatant incubation), to explore tumoricidal activity (Figure 4A). As such, all of these macrophages without WGP demonstrated similar tumor burdens, as indicated by fluorescent intensity after 24 h incubation (Figure 4B). Meanwhile, WGP enhanced tumoricidal activity prominently in M0 and M2 but less profoundly in TAM, without any beneficial effects on M1 macrophages (Figure 4B). Regarding macrophage polarization genes, M1 upregulated *IL-1β* and *iNOS* and M2 increased *TGF-β, Fizz-1*, and *Arg-1*, while TAM mildly enhanced *iNOS, TGF-β, Fizz-1,* and *Arg-1* (Figure 4C–H). With WGP, *IL-1β* and *iNOS* were upregulated in all types of macrophages, except non-elevated *iNOS* in M1 (Figure 4D,E). On the other hand, WGP downregulated *TGF-β, Fizz-1,* and *Arg-1* in M2 and reduced *TGF-β* and *Arg-1* in TAM but upregulated *TGF-β* in M1 (Figure 4F–G), supporting WGP-induced M1 macrophage polarization with pro-inflammatory properties. For inflammatory mediators, M1 increased higher supernatant TNF-α, IL-6, and IL-10, with the highest upregulation of *TNF-α* and *IL-6* compared to other macrophages, while *IL-10* was enhanced in all types of macrophages compared with the M0 control (Figure 5A–F). Additionally, the upregulation of *TLR-4* and *NFκB* was demonstrated only in M1, while increased *Dectin-1* was found in all types of macrophages when compared with M0 control (Figure 5G–J). With WGP, there were only upregulations of *TNF-α* and *Dectin-1,* but not other parameters, in M0, M2, and TAM, without any effects on M1 (Figure 5A–J), perhaps in correlation with the neutral effect of WGP on M1 in macrophage tumoricidal activity (Figure 4C). In the comparison to the M0 control, all interventions (LPS, IL-4, tumor supernatant, and WGP) upregulated *Dectin-1* and the elevated *Dectin-1* from IL-4 and tumor supernatant induction, but not LPS, could be enhanced by WGP (Figure 5I). Because of the recognition of WGP by Dectin-1, the enhanced *Dectin-1* (Figure 5I) might be responsible for the elevated tumoricidal activity of macrophages (Figure 4C).

### 2.3. Cell Energy Status in Different Types of Macrophages and the Impact of WGP

Because of the well-known association between cell energy status and macrophage function (prominent glycolysis in M1 pro-inflammatory and profound mitochondrial activity in M2 alternative macrophage polarization) [23] and the correlation between Dectin-1 activation and cell energy status [24], WPG might alter the cell energy of macrophages (Figure 6A–F). Without WGP, the mitochondrial activity in M1 (LPS-induced macrophages) was lower than in the M0 control, while the activity in M2 (IL-4 activation) and TAM (supernatant tumor induction) were not significantly different to the M0 control, as indicated by the oxygen consumption rate (OCR) (Figure 6A upper) and the respiratory parameters (maximal respiration and respiratory reserve) (Figure 6E). For glycolysis activity (without WGP), M1 and M2 (but not TAM) demonstrated a higher extracellular acidification rate (ECAR) than the M0 control, as indicated by the value in the graph (Figure 6A lower); however, the glycolysis activity as calculated by the area under the curve (AUC) of ECAR of all the activated macrophages (M1, M2, and TAM) was higher than that of the M0 control (Figure 6F). With WGP (Figure 6B), there was an increase in mitochondrial functions (maximal respiration and respiratory reserve) without an alteration in glycolysis activity (Figure 6E,F). In M2 and TAM (Figure 6C,D), WGP enhanced both mitochondrial and glycolysis activities when compared with M2 or TAM alone (Figure 6E,F). The enhanced cell energy status in M2 and TAM by WGP (Figure 6F–L) might be responsible for the enhanced tumoricidal activity of M2 and TAM against the MC38 colon cancer cell line (Figure 4C).

## 3. Discussion

Oral administration of *Saccharomyces cerevisiae* reduced the growth of colon cancers in cGAS-deficient (cGAS-/-) mice in the models using azoxymethane (AOM) induction and subcutaneous injection of cancer cells, possibly through energy enhancement in macrophages through whole glucan particle (WGP)-induced Dectin-1.

### 3.1. Impacts of Environemental Factors and Genetic Susceptibility in Spontaneous Colon Cancer in cGAS-/- Mice and Saccharomyces cerevisiae Administration

Although spontaneous colon cancer activation by AOM is frequently used in wildtype (WT) mice [25,26,27], the protocol could not induce cancer in our WT mice and activated only some cGAS-/- mice, possibly due to the difference in the gut microbiota of mice in the different animal facility environments. Repeated AOM in several doses and/or increased doses of AOM might increase cancer lesions in WT mice [28]. However, our proof of concept experiments supported the importance of cGAS in cancer development [29]. A single administration of a hydrocarbon compound with chronic intestinal inflammation by dextran sulfate, a substance which directly affects the enterocyte tight junction [30], together with a lack of dsDNA recognition by cGAS deficiency, is an example of colon cancer development that might also be possible in humans. Indeed, chronic intestinal inflammation without any intestinal symptoms is possible in several situations, as it can be detected through an enhanced translocation of pathogen molecules from the gut into the blood circulation (gut barrier defect, gut leakage or leaky gut) [31]. Examples of asymptomatic chronic intestinal inflammation are obesity (systemic inflammation-induced leaky gut) [32,33,34], uremia (uremic toxin-induced intestinal damage) [35,36], iron overload (enterocyte iron toxicity) [37], autoimmune diseases (circulating-immune complex deposition in the gut) [38,39,40], prolong oral administration of some drugs [41], and dysbiosis from several diseases [42].

The increased susceptibility to AOM-induced colon cancer in cGAS-/- mice is possibly not only a result of the defect in dsDNA recognition but might also from gut dysbiosis from cGAS deficiency. With the defects in cGAS, there might be a defect in immune responses against several intracellular bacteria, including obligate intracellular bacteria (*Coxiella burnetti*, *Chlamydia, Anaplasma, Ehrlichia, Rickettsia, Orientia,* and *Mycoplasma*) [43] and non-obligate intracellular bacteria (*Salmonella, Listeria, Brucella, Rickettsia,* and *Legionella*) [44], that are naturally pass through the macrophage cell membrane and activate the cytosolic cGAS receptor [45]. Indeed, the dissimilarity in fecal microbiome analysis between WT and cGAS-/- mice was demonstrated by different distances from the axis in principal coordinate analysis (PCoA) and different representative organisms in each group from linear discriminant analysis (LDA). As such, cGAS-/- mice demonstrated a higher abundance of *Rikennella* (Gram-negative anaerobic bacilli) [46] and *Turricibator* (Gram-positive anaerobic bacilli) than WT mice. However, these bacteria are difficult to culture and the data on cancer association is still unknown. Nevertheless, there was no colon cancer in *S. cerevisiae*-administered cGAS-/- mice, suggesting a beneficial effect associated with yeast probiotics against cancers, as previously mentioned [1]. Despite the several anti-cancer mechanisms of the yeast cells, a possible effect of *S. cerevisiae* on the attenuation of gut dysbiosis was indicated by an increase in some beneficial bacteria against cancers, including *Lactobacilli* and *Akkermansia* [47,48], in cGAS-/- feces. Hence, the combination probiotics using *S. cerevisiae* with other bacteria is of interest concerning cancer prevention.

### 3.2. Beta-Glucan from the Cell Wall of Saccharomyces cerevisiae Attenuated Subcutaneous Tumor Growth Partly through Dectin-1-Mediated Cell Energy Enhancement in Macrophages

Additionally, *S. cerevisiae* might manipulate macrophage activation by the beta-glucan component of the cell wall. Despite the deficiency in the cGAS receptor, the injection of WGP (a commercially available *S. cerevisiae* glucan) or the in-house extract of glucan inhibited the growth of subcutaneous tumors in cGAS-/- mice suggested that non-cGAS-mediated anti-cancer mechanisms are associated with beta-glucan. Notably, the reduced burdens of cancer cells in glucan-administered mice resulted in lower serum IL-1β and IL-6 (the cytokines that might be associated with tumor growth) [49]. Although glucan might be the main component of the yeast extract, the difference in serum cytokines after the administration of yeast extract versus the commercially available WGP (higher serum IL-1β with lower IL-6 after WGP) implied a possible contamination in the in-house yeast extract procedure. Due to the importance of macrophages in inflammatory responses (M1 and M2 of pro- and anti-inflammation, respectively) and in cancers (M2-liked tumor-associated macrophages; TAM), several subtypes of macrophages were tested. All the subtypes of macrophages here (M1, M2, and TAM) demonstrated a similar tumoricidal activity to control M0, as indicated by the reduction in fluorescent activities despite the well-known prominent tumoricidal activity of M1 compared to TAM [50], implying the importance of cytotoxic T cells (Tc) in anti-cancer activity. Indeed, TAM and M1 promote and inhibit cancers, respectively, partly through the blockage and facilitation of Tc [51]. On the other hand, with WGP, there was an enhanced tumoricidal activity with the upregulated genes of M1 polarization in non-LPS activated macrophages (M0, M2, and TAM), despite a lessor tumoricidal activity in WGP-activated TAM among all groups. Although WGP upregulated *IL-1β* in M1, WGP could not enhance tumoricidal activity, implying non-cytokine-dependent anti-tumor mechanisms [52].

Because WGP activates macrophages through Dectin-1 and other inflammatory signals [53,54], several genes were explored. Indeed, all activators (LPS, IL-4, WGP, and tumor supernatant) upregulated *Dectin-1*, but not *TLR-2* and *TLR-4*, highlighted an enhancement of Dectin-1 on the non-specific activations of macrophages, perhaps as a preparation for the possible following activations [24,55,56,57]. Although TLR-2 and TLR-4 might possibly recognize WGP [58], Dectin-1 might be the most important pattern recognition receptor for WGP. In M1 macrophages, WGP could not up-regulate *Dectin-1*, in parallel with a failure in enhanced tumoricidal activity. Meanwhile, WGP enhanced *Dectin-1*, together with the increased tumoricidal activity of M0, M2, and TAM. Hence, Dectin-1 facilitation might be responsible for macrophage tumoricidal activity, supporting previous publications [59,60]. Due to the profound potency in terms of the inflammatory activator and cell energy alteration of LPS [10] when compared with IL-4 and tumor supernatant, the limited *Dectin-1* upregulation in WGP-activated M1 might be because of the lack of cell energy after LPS stimulation. Indeed, the extracellular analysis demonstrated low mitochondrial activity with high glycolysis in both M1 and TAM when compared with M0, similar to a previous publication [61,62], but the more prominent mitochondrial defect, especially in the respiratory reserve, was demonstrated more clearly in M1 than TAM. In parallel, considering the preserve of mitochondrial activity, the energy status of M2 was similar to TAM [63,64] when compared with the M0 control. Nevertheless, WGP improved the cell energy status (both in terms of mitochondria and glycolysis) in all subtypes of macrophages. From these results, the failure of M1 in *Dectin-1* upregulation and tumoricidal enhancement might be due to the profound defect in mitochondria after LPS stimulation [65,66], which might be beyond the point that could be significantly improved by WGP. Interestingly, the impact of WGP on the induction of tumoricidal activity in TAM might be responsible for tumor attenuation in AOM-mediated colon cancer and subcutaneous tumor injection, which will be beneficial for protection against colon cancer. Although the influence of mitochondrial activity in cancer cells is well-known [67,68,69], data on the impact of the mitochondria of immune cells and Dectin-1 is less documented. We hypothesize that upregulated *Dectin-1* and mitochondrial improvement in terms of macrophages might be associated with macrophage anti-cancer activities, as presented in Figure 7. Notably, the tests concerning macrophage stimulation and cell energy manipulation using the in-house yeast extract were not performed here due to an awareness of the standard of the preparation procedures. However, yeast extract might be an economical source of glucan for real clinical settings in some developing countries. More studies on these topics would be of interest.

## 4. Materials and Methods

### 4.1. Animal and Animal Models

Animal care and use protocol based upon the National Institutes of Health (NIH), USA was approved by the Institutional Animal Care and Use Committee- of the Faculty of Medicine, Chulalongkorn University, Bangkok, Thailand. Wild-type male 8-wk-old mice on C57BL/6j background were purchased from Nomura Siam International (Pathumwan, Bangkok, Thailand). Likewise, cyclic GMP-AMP synthase (cGAS) deficient mice in C57BL/6J background (cGAS-/-) were kindly provided by Paludan (Aarhus University, Aarhus, Denmark), and only male 8-week-old mice were used [70]. The mice were housed in standard clear plastic cages (3–5 mice per cage) with free access to water and food (SmartHeart Rodent, Perfect companion pet care, Bangkok, Thailand), air change ration at 15 air changes per h, and a light/dark cycle of 12:12 h in 22 ± 2 °C with 50 ± 10% relative humidity. A chronic inflammation-driven colon cancer model by intraperitoneal administration of azoxymethane (AOM) at 10 mg/kg body weight with 3 cycles of 1 week in duration of 2.5% (*w*/*v*) dextran sulfate solution (DSS) solution, with a 2 week resting period, was performed following a previous protocol [25,26]. Then, 1 × 10^8^ CFU of *S. cerevisiae* (ATCC 1171) (The American Type Culture Collection, Manassas, VA, USA) in 500 μL normal saline solution (NSS) of NSS alone was administered by oral gavage every other day from the 12th to the 24th week of the experiment (Figure 1A) using 18 gauge feeding tubes 1.5 inches in length with a rounded tip mouse oral gavage needle (Sigma-Aldrich, St. Louis, MO, USA). All mice were sacrificed in the 24th week of the experiment by cardiac puncture under isoflurane anesthesia, and samples were collected (small and large bowels, blood, and feces). Notably, *S. cerevisiae* was prepared on Sabouraud dextrose agar (SDA) (Oxiod, Basingstoke, Hampshire, UK) for 24 h at 35 °C before resuspension in NSS to determine the abundance using the Mcfarland method (0.5 Mcfarland for approximately 1 x 10^8^ cell/mL).

To explore the anti-cancer mechanism of *S. cerevisiae*, the subcutaneous injection of a colon cancer cell line (MC38) (C57BL6 murine colon adenocarcinoma cells) (Kerafast, Boston, MA, USA) was performed. As such, MC38 at 1 x 10^5^ cells per mice in 100 µL of culture media, using Dulbecco’s Modified Eagle Medium (DMEM) with 10% fetal bovine serum (FBS), was subcutaneously injected into the left flank area following a previous publication [71]. At 2 weeks post-injection, 10 mg of whole glucan particle (WGP), the soluble (1→3)/(1→6)-β-Glucan) extracted from *S. cerevisiae* (InvivoGen, San Diego, CA, USA) in 100 μL of normal saline solution (NSS) or NSS alone or the crude extract of *S. cerevisiae* (yeast extract) were intralesionally injected once daily from the 2nd to the 4th week of experiments. The crude extract *S. cerevisiae* was performed following a previous protocol [72]. Briefly, a heating process (50 °C for 24 h) was used to prepare the autolyzed yeast before mixing with 1 M NaOH at 80 °C in a stirrer for 2 h, dissolving the pellets with distillation water resuspension, centrifuged again with dissolving in 1 M HCl at 80 °C in a stirrer for 2 h, and centrifuged to retrieve the yeast extract pellets. The pellets were washed with distillation water 3 times, dried in a hot air oven (60 °C), and kept at 4 °C before use. Then, 10 mg of the yeast extract was dissolved by 1 M NaOH before pH neutralization and injected into the tumor. Tumor volume was observed as previously mentioned [71], and all mice were sacrificed at the 4^th^ week of the experiment by cardiac puncture under isoflurane anesthesia, and samples were collected (tumors and blood). Serum cytokines were determined by enzyme-linked immunosorbent assays (ELISAs) (Invitrogen, Waltham, MA, USA).

### 4.2. Fecal Microbiome Analysis

Using the DNA from each mouse’s feces (0.25 g per mouse), a fecal microbiota study was carried out following the protocols described in earlier publications [35,42,73,74]. In brief, the power DNA isolation assay (MoBio, Carlsbad, CA, USA) and agarose gel electrophoresis with nanodrop spectrophotometry were utilized for total DNA extraction and metagenomic DNA quality determination, respectively. The universal prokaryotic primers 515F (forward) and 806R (reverse), 5’-GTGCCAGCMGCCGGTAA-3’ and 5’-GGACTACHVGGGTWTCTAAT-3’, respectively, and a 16S rRNA V4 library (appended 50 Illumina adapter and 30 Golay barcode sequences) were used. Each sample (240 ng) was put on the MiSeq300 sequencing platform (Illumina, San Diego, CA, USA) with Mothur’s standard quality screening operating procedures. Aligned and assigned taxa (operational taxonomic units [OTUs]) based on default parameters were employed in the MiSeq platform [36,75,76,77].

### 4.3. Macrophage Experiments and Fluorescent Labelling Cancer Cells

To explore the impacts of macrophages and yeasts on tumoricidal activities, in vitro experiments were conducted. Bone marrow (BM)-derived macrophage preparation from mouse femurs using supplemented DMEM with a 20% conditioned medium of the L929 cell line (ATCC CCL-1), the fibroblasts used as a source of macrophage colony-stimulating factor (M-CSF), in 5% CO2 humidified incubator at 37 °C for 7 days before harvesting with cold PBS was conducted [37,54,70,78]. The macrophages at 5 x 10^4^ cells/well in DMEM supplemented with 10% heat-inactivated fetal bovine serum (FBS) and Penicillin-Streptomycin (Thermo Fisher Scientific, Waltham, MA, USA) were incubated in 5% carbon dioxide (CO2) at 37 °C for 24 h before being treated for another 24 h by lipopolysaccharide (LPS) (*Escherichia coli* 026: B6; Sigma-Aldrich, St. Louis, MO, USA) (100 ng/mL) or IL-4 (20 ng/mL) to induce M1 pro-inflammatory and M2 alternative (anti-inflammatory) macrophage polarization, respectively [79], or DMEM alone (M0). In parallel, tumor-conditioned media from the MC38 cell line (with IL-4 cocktails) were used to activate tumor associate macrophages (TAM) following a previous publication [80]. For tumor-conditioned media, MC38 cells at 4 × 10^6^ cells in 8 mL in modified DMEM were centrifuged to remove the suspended cells [80]. All subtypes of macrophages (M0, M1, M2, and TAM) were incubated at 1 × 10^5^ cells/well with 1 × 10^5^ MC-38 cancer calls that were previously labeled with Carboxyfluorescein diacetate succinimidyl ester (CFDA-SE) (Sigma-Aldrich, St. Louis, MO, USA) with WGP (100 μg/mL) or DMEM control for 24 h before sample collection (cells and supernatant). The intensity of fluorescent cells using a fluorescent microscope was determined by ImageJ (National Institutes of Health, MD, USA). For fluorescent labeling, CFDA-SE (20 μM in PBS) was incubated with 1 × 10^5^ cancer calls for 30 min at 37 °C before the gentle removal of the buffer, with a further 15 min of incubation and cell collection, according to the manufacturer’s protocol. Notably, the passively diffused CFDA-SE in the cytoplasm was cleaved by intracellular-esterase to form the fluorescent activity.

### 4.4. Gene Expression, Supernatant Cytokines, and Extracellular Flux Analysis

The influence of WGP or DMEM against different macrophages (M0, M1, M2, and TAM) was evaluated after 24 h incubation, as determined by the expression of M1 macrophage polarization (*IL-1β* and *iNOS*), M2 polarization (*TGF-β, Arginase-1,* and *Fizz-1*), inflammatory cytokines (*TNF-α, IL-6,* and *IL-10*), pattern recognition receptors (*TLR-2, TLR-4,* and *Dectin-1*), and *NFκB* downstream signaling using the primers listed in Table 1. In parallel, supernatant cytokines (TNF-α, IL-6, and IL-10) were measured by ELISAs (Invitrogen, Waltham, MA, USA). To determine an alteration in macrophage cell energy, extracellular flux analysis was conducted using Seahorse XFp Analyzers (Agilent, Santa Clara, CA, USA) with the oxygen consumption rate (OCR) and extracellular acidification rate (ECAR) representing mitochondrial function (respiration) and glycolysis activity, respectively, as previously described [41,81,82]. In the OCR evaluation, the stimulated macrophages at 1 × 10^5^ cells/well were incubated for 1 h in Seahorse media (DMEM complemented with glucose, pyruvate, and L-glutamine) (Agilent, Santa Clara, CA, USA; 103575–100) before activation by different metabolic interference compounds such as oligomycin, carbonyl cyanide-4-(trifluoromethoxy)-phenylhydrazone (FCCP), and rotenone/antimycin A. Meanwhile, the respiratory data for mitochondrial function were analyzed by Seahorse Wave 2.6 software based on the following equations: maximal respiration = OCR between FCCP and rotenone/antimycin A; OCR after rotenone/antimycin A and respiratory reserve = OCR between FCCP and rotenone/antimycin A; OCR before oligomycin. In parallel, glycolysis stress tests were calculated from the mitochondrial stress test using the wave program Seahorse XF Analyzers (Agilent, Santa Clara, CA, USA) and demonstrated by the area under the curve (AUC) of the ECAR graph, as calculated by the trapezoidal rule [83].

### 4.5. Statistical Analysis

All data were analyzed by the Statistical Package for Social Sciences software (SPSS 22.0, SPSS Inc., Chicago, IL, USA) and Graph Pad Prism version 7.0 software (La Jolla, CA, USA). Results were presented as mean ± standard error (SE). The differences between multiple groups were examined for statistical significance by one-way analysis of variance (ANOVA) with Tukey’s analysis. The survival analysis and time-point data were determined by the log-rank test and repeated measures ANOVA, respectively. A *p*-value < 0.05 was considered statistically significant.

## 5. Conclusions

In conclusion, *S. cerevisiae* attenuated colon cancer in AOM-induction and subcutaneous injection models through dysbiosis attenuation and immune modulation by beta-glucan from the cell wall. The *S. cerevisiae* glucan upregulated *Dectin-1* and enhanced macrophage tumoricidal activities, partly through the improved macrophage cell energy status. Our data support the use of *S. cerevisiae* or beta-glucan for colon cancer prevention.

## Figures and Tables

**Figure 1 ijms-23-10951-f001:**
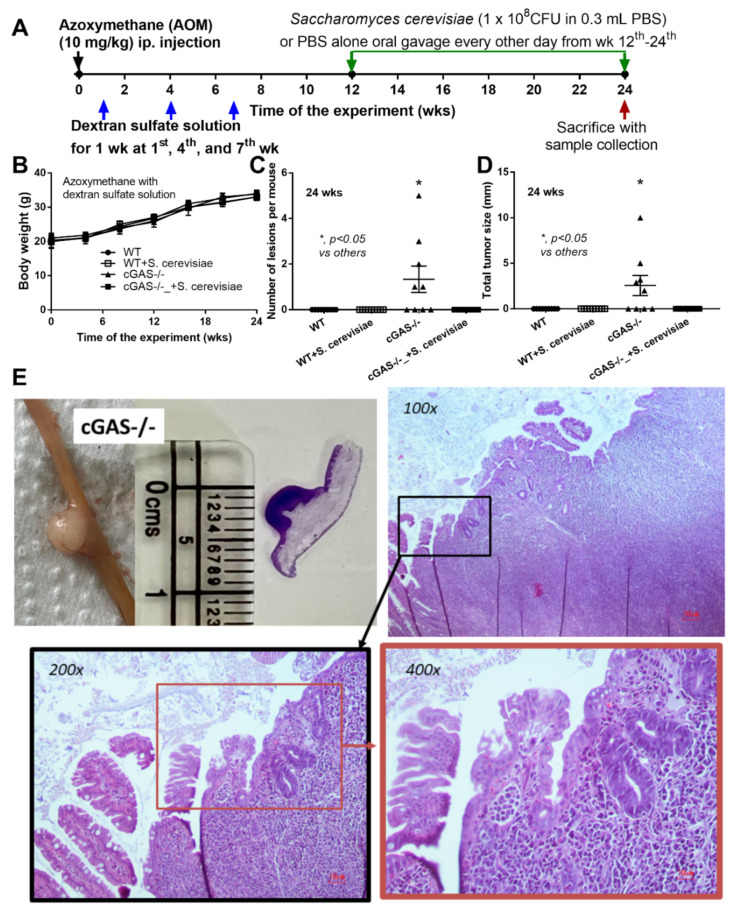
Schema of the experiments of azoxymethane (AOM) with dextran sulfate solution (DSS)-induced colon cancer and oral gavage of *Saccharomyces cerevisiae* or phosphate buffer solution (PBS) control in cGAS-deficient (cGAS-/-) and wild-type (WT) mice (**A**); characteristics of the mice as indicated by body weight (**B**), number of lesions (**C**), total tumor size (diameter of each polyp multiplied by the number of polyps displaying this diameter) (**D**); a representation of the tumor by direct visual observation (from mice and histological slide with a scale in centimeters) (**E**, left upper) and Hematoxylin and Eosin (H&E) staining from a polyp-liked lesion indicating the monomorphic bizarre cell morphology of the cancer lesion, with the original magnification at 100x–400x (**E**, histological pictures) (*n* = 9–10/group) (ip., intraperitoneal injection). The analysis of time-point data (**B**) and the multiple groups (**C**,**D**) were determined by repeated measures ANOVA and one-way analysis of variance (ANOVA) with Tukey’s analysis, respectively. The exact *p* values for (**C**,**D**) are demonstrated in Appendix A.

**Figure 2 ijms-23-10951-f002:**
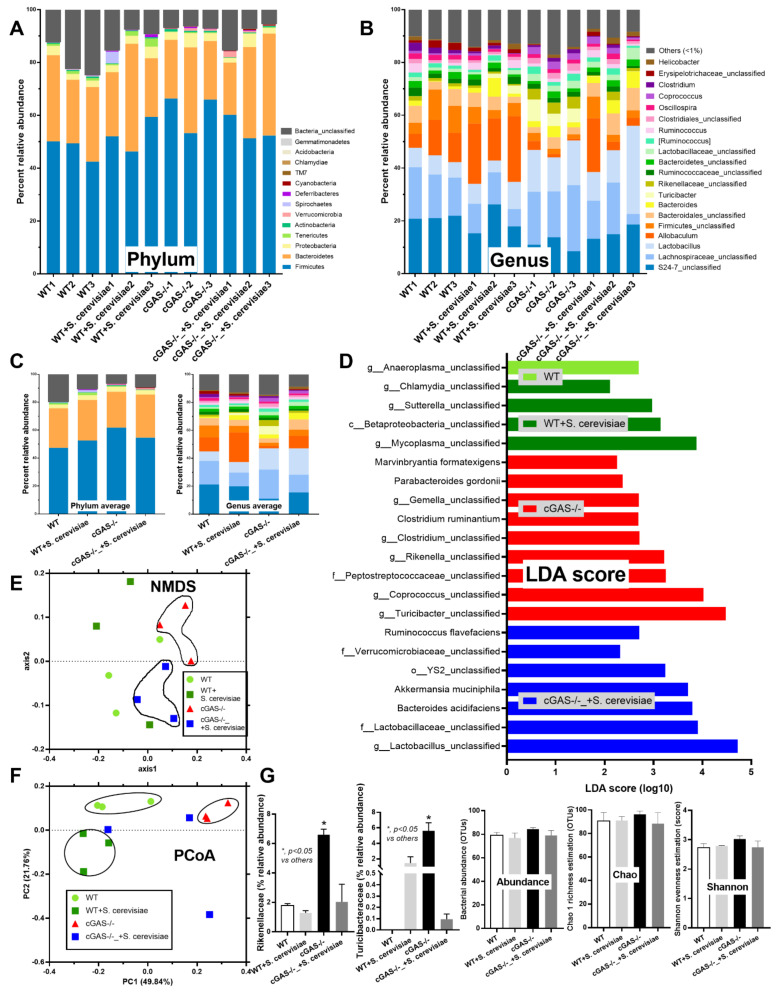
Fecal microbiome analysis from azoxymethane (AOM) with dextran sulfate solution (DSS)-induced colon cancer and oral gavage of phosphate buffer solution (PBS) control or *Saccharomyces cerevisiae* (*S. cerevisiae*) in cGAS-deficient (cGAS-/-) and wild-type (WT) mice as indicated by the abundance of fecal bacteria in phylum and genus level with the average value (**A**–**C**); the possible unique bacteria in each group using linear discriminant analysis (LDA score) (**D**); the dissimilarity among each group by distance from the axis with non-metric multidimensional scaling (NMDS) (**E**); the principal coordinate analysis (PCoA) of the community structure using ThetaYC distances (**F**); and a graph demonstration of significant bacteria, with the total bacterial abundance in operational taxonomic units (OTUs) and alpha-diversity analysis (Chao-1 and Shannon score) (*n* = 3/group). *, *p* < 0.05 vs. other groups. The analysis of multiple groups (**G**) were determined by one-way analysis of variance (ANOVA) with Tukey’s analysis. The exact *p* values were demonstrated in Appendix A.

**Figure 3 ijms-23-10951-f003:**
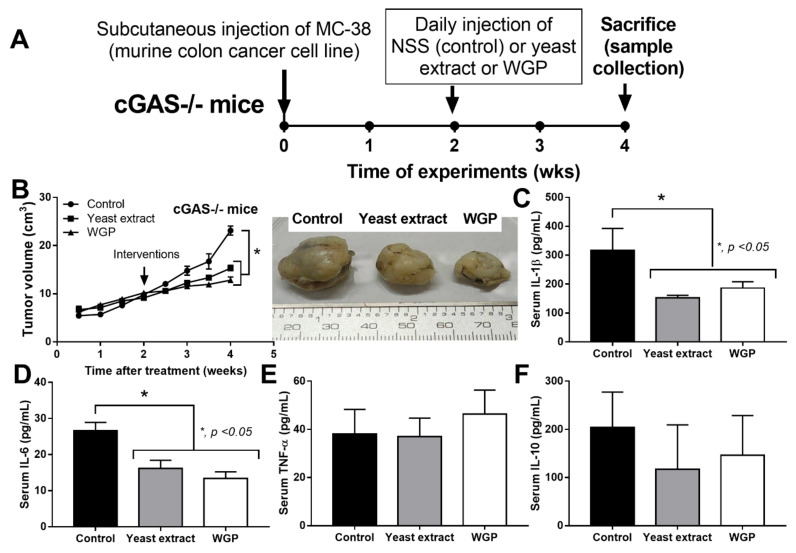
Schema of the experiments of subcutaneous injection of a murine colon cancer cell line (MC38) in cGAS-deficient (cGAS-/-) mice with daily intralesional injection by normal saline solution (NSS) control, crude extract of *Saccharomyces cerevisiae* (yeast extract), or whole glucan particle (WGP) starting from the 2nd to 4th week of experiments (**A**) with characteristics of the mice as indicated by tumor volume with the representative pictures of the excised tumors (**B**) and serum cytokines (IL-1β, IL-6, TNF-α, and IL-10) (**C**–**F**) (*n* = 7–8/group). **, p < 0.05.* The analysis of time-point data (**B**) and the multiple groups (**C**–**F**) were determined by repeated measures ANOVA and one-way analysis of variance (ANOVA) with Tukey’s analysis, respectively. The exact *p* values for (**A**,**C**,**D**) are demonstrated in Appendix A.

**Figure 4 ijms-23-10951-f004:**
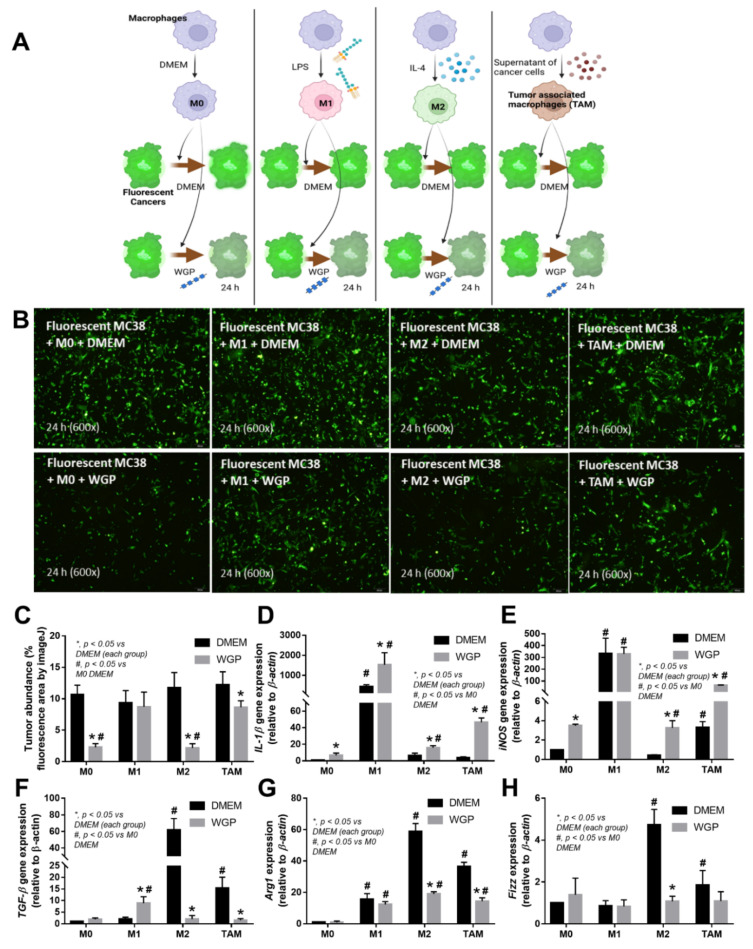
Schema of the in vitro experiments (**A**) with the incubation of fluorescence-stained colon cancer cell line (MC38) with different bone marrow-derived macrophages of the wild-type mice, including M0 (incubation with DMEM control media), M1 (LPS activation), M2 (IL-4 stimulation), and tumor-associated macrophages (TAM; using MC38 supernatant), together with whole glucan particle (WGP) or DMEM media control (**A**), with the characteristics of DMEM- or WGP-activated experiments, as indicated by tumor burdens (fluorescent intensity with the representative pictures) (**B**,**C**), genes of M1 macrophage polarization (pro-inflammation) (*IL-1β* and *iNOS*) (**D**,**E**), and M2 polarization (anti-inflammation) (*TGF-β, Arg-1,* and *Fizz*) (**F**–**H**). Independent triplicate experiments were performed. *, *p* < 0.05 vs. DMEM in each group; #, *p* < 0.05 vs. M0 (DMEM). The picture was generated by BioRender.com.

**Figure 5 ijms-23-10951-f005:**
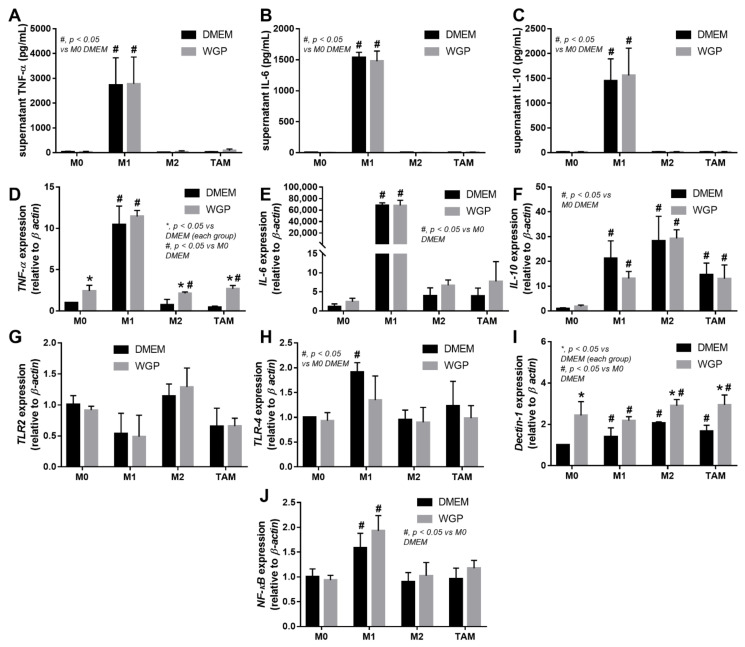
The characteristics of the in vitro experiments with the incubation of fluorescence-stained colon cancer cell line (MC38) with different bone marrow-derived macrophages of the wild-type mice, including M0 (incubation with DMEM control media), M1 (LPS activation), M2 (IL-4 stimulation), and tumor-associated macrophages (TAM; using MC38 supernatant), together with whole glucan particle (WGP) or DMEM media control, as indicated by supernatant pro-inflammatory cytokines (**A**–**C**) and gene expression (*TNF-α, IL-6,* and *IL-10*) (**D**–**F**) with inflammatory signals (*TLR-2, TLR-4, Dectin-1,* and *NFκB*) (**G**–**J**). Independent triplicate experiments were performed. *, *p* < 0.05 vs. DMEM in each group; #, *p* < 0.05 vs. M0 (DMEM).

**Figure 6 ijms-23-10951-f006:**
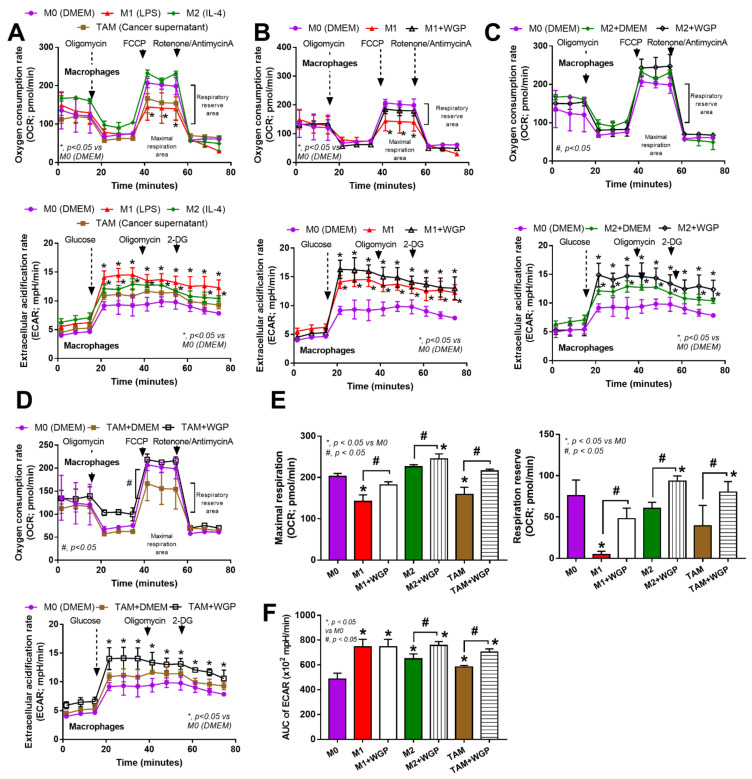
The characteristics of cell energy of bone marrow-derived macrophages from wild-type mice after incubation with DMEM control media (M0), LPS (M1), IL-4 (M2) and the supernatant of cancer cell line (MC38) (tumor-associated macrophages; TAM) with or without the whole glucan particle (WGP), as indicated by graphs of extracellular flux analysis for mitochondrial function (oxygen consumption rate; OCR) and glycolysis activity (extracellular acidification rate; ECAR) (**A**–**D**), with a graph presentation of cell energy parameters, including mitochondrial functions (maximal respiration and respiratory reserve) (**E**) and area under the curve (AUC) of ECAR (glycolysis activity) (**F**). Independent triplicate experiments were performed. *, *p* < 0.05 vs. M0; #, *p* < 0.05.

**Figure 7 ijms-23-10951-f007:**
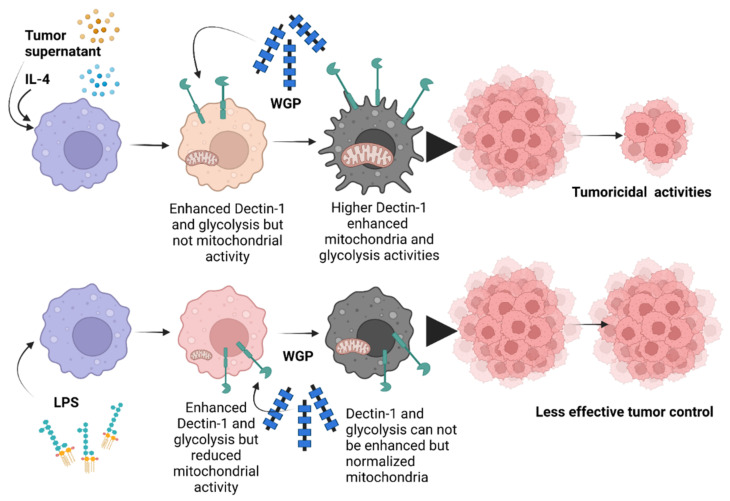
The proposed working hypothesis demonstrates the role of whole glucan particles (WGPs) in the enhancement of the tumoricidal activities of macrophages. While all activators, including the tumor microenvironment (using tumor supernatant for inducing tumor-associated macrophages; TAM), IL-4 (M2), and lipopolysaccharide (LPS) (M1), had an effect on upregulated *Dectin-1* and increased glycolysis, LPS more prominently reduced mitochondrial activities when compared with tumor supernatant and IL-4. Then, *Dectin-1* was further upregulated by WGP in non-LPS-activated macrophages (M2 and TAM) and further upregulated by WGP stimulation, leading to an elevation in glycolysis and mitochondrial activities partly through WGP-Dectin-1 activation. In parallel, the WGP-improved glycolysis was more prominent in M2 and TAM than in M1 (possibly due to the already high glycolysis present in M1) and is possibly correlated with enhanced tumoricidal activity in M2 and TAM, implying a glycolysis-dependent tumoricidal effect. The picture was created by BioRender.com and is available online: https://app.biorender.com/ (accessed on 7 August 2022).

**Table 1 ijms-23-10951-t001:** List of primers used in the study.

Primers	Forward	Reverse
Tumor necrosis factor-alpha (*TNF-α*)	5′ -CCTCACACTCAGATCATCTTCTC- 3′	5′ -AGATCCATGCCGTTGGCCAG- 3′
Interleukin-6 (*IL-6*)	5′ -TACCACTTCACAAGTCGGAGGc- 3′	5′ -CTGCAAGTGCATCATCGTTGTTC- 3′
Interleukin-10 (*IL-10*)	5′ -GCTCTTACTGACTGGCATGAG- 3′	5′ -CGCAGCTCTAGGAGCATGTG- 3′
Inducible nitric oxide synthase (*iNOS*)	5′ -ACCCACATCTGGCAGAATGAG- 3′	5′ -AGCCATGACCTTTCGCATTAG- 3′
Interleukin-1ß (*IL-1ß*)	5′ -GAAATGCCACCTTTTGACAGTG- 3′	5′ -TGGATGCTCTCATCAGGACAG- 3′
Arginase-1 (*Arg-1*)	5′ -CTTGGCTTGCTTCGGAACTC- 3′	5′ -GGAGAAGGCGTTTGCTTAGTTC- 3′
Transforming Growth Factor-β (*TGF-β*)	5′ -CAGAGCTGCGCTTGCAGAG- 3′	5′ -GTCAGCAGCCGGTTACCAAG- 3′
Resistin-like molecule-α (*FIZZ-1*)	5′ -GCCAGGTCCTGGAACCTTTC- 3′	5′ -GGAGCAGGGAGATGCAGATGA- 3′
Nuclear factor-κB (*NF-κB*)	5′ -CTTCCTCAGCCATGGTACCTCT- 3′	5′ -CAAGTCTTCATCAGCATCAAACTG- 3′
Toll like receptor-2 (*TLR-2*)	5′ -ACAGCAAGGTCTTCCTGGTTCC- 3′	5′ -GCTCCCTTACAGGCTGAGTTCT- 3′
Toll like receptor-4 (*TLR-4*)	5′ -GGCAGCAGGTGGAATTGTAT- 3′	5′ -AGGCCCCAGAGTTTTGTTCT- 3′
*Dectin-1*	5′ -TCCCGCAATCAGAGTGAAG- 3′	5′ -GTGCAGTAAGCTTTCCTGGG- 3′

## Data Availability

Data are contained within the article.

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
