# Peer review of "Beta-Glucan from S. cerevisiae Protected AOM-Induced Colon Cancer in cGAS-Deficient Mice Partly through Dectin-1-Manipulated Macrophage Cell Energy"

_ijms, 2022, doi:10.3390/ijms231810951_

Round 1

Reviewer 1 Report

Very interesting study, a well-planned in vivo and in vitro experiment, good discussion and correct conclusions.I have only a few comments as below:

Keywords: dysbiosis and Dectin-1 should be added

Introduction: (1) ref. 1 - is this the right quote?

Results: Fig. 2E - more details for histological photos should be added

Materials and Methods: (1) more details for animal's experiment (environmental conditions, i.e. temperature, humidity, air change ratio, gavage description, aimal's feed basal content) should be added; (2) which "organs" were collected? should be given.

Author Response

Keywords: dysbiosis and Dectin-1 should be added

Introduction: (1) ref. 1 - is this the right quote?

ANS: We thank the reviewer for the comment and add a review on colon cancer in that position. 

Results: Fig. 1E - more details for histological photos should be added

ANS: We thank the reviewer for the comment and explain more in the figure legend. 

Materials and Methods: (1) more details for the animal's experiment (environmental conditions, i.e. temperature, humidity, air change ratio, gavage description, animal's feed basal content) should be added;

ANS: We thank the reviewer for the comment and add more data. 

(2) which "organs" were collected? should be given.

ANS: We thank the reviewer for the comment and explain more (small and large bowels). 

Reviewer 2 Report

In this study, the authors claimed that  S. cerevisiae attenuated colon cancer in AOM-induction and graft mouse models through mainly the effect of cell wall B-glucan on the immune modulation and Dectin-1 which  upregulated the macrophage tumoricidal activities.

Major concerns

1- Figure 1:  

a) Figure 1D and IC, please include the exact value of p value and the method used for analysis in the legend

b) Figure 1D: the authors mentioned in the legend (total tumor size (diameter of each lesion multiplied by numbers of the lesion)), how can this calculated?. I guess the author means the diameter of each polyp x number of polyps displaying this diameter, not the total no of polyps. Also It is surprised that some polyps have  the same diameter???

c) Histology does not give any scientific information, why the authors added it?  Is it from non-involved or polyp region? Also are the magnification written correct? I guess they should be 10x, 20x, and 40x, please revise. What is the image scale?

Figure 2)

a)Same comment regarding the statistics. Please add the exact value, method used in the legend.

b) Since there was no significant difference between microbiome abundance/ diversity in the different group, I think the conclusion that  S. cerevisiae could partly affect the microbial dysbiosis is not solid. please rephrase

Figure 3:

a)Same comment regarding the statistics. Please add the exact value, method used in the legend.

b) WGP produced significant amount of IL1-B than yeast extract, while yeast produced more IL-6, any verification for this point?

Figure 4 and 5

a) why the authors did not do similar experiments with yeast extract and compare between WGP and yeast extract in vitro?

Author Response

1- Figure 1:  

  1. Figure 1D and IC, please include the exact value of p value and the method used for analysis in the legend

ANS: We thank the reviewer for the comment and added it accordingly. The exact p values were added in the supplement Table.

  1. b) Figure 1D: the authors mentioned in the legend (total tumor size (diameter of each lesion multiplied by numbers of the lesion)), how can this calculated?. I guess the author means the diameter of each polyp x number of polyps displaying this diameter, not the total no of polyps. Also It is surprised that some polyps have  the same diameter???

ANS: We thank the reviewer for this important comment and corrected it accordingly. The too crude estimation in the former figure is replaced by the more solid measurement values. 

  1. c) Histology does not give any scientific information, why the authors added it?  Is it from non-involved or polyp region? Also are the magnification written correct? I guess they should be 10x, 20x, and 40x, please revise. What is the image scale?

ANS: We thank the reviewer for the comment. We would like to demonstrate the characteristics of cancer from the model (monomorphic bizarre nuclei) from the cancer (polyp-liked lesion). The magnification is 100x due to the 10x from the eyepiece and 10x (20x or 40x) from the objective lens.

Figure 2)

a)Same comment regarding the statistics. Please add the exact value, method used in the legend.

ANS: We thank the reviewer for the comment and added it accordingly. The exact p values were added in the supplement Table.

  1. Since there was no significant difference between microbiome abundance/ diversity in the different group, I think the conclusion that  cerevisiae could partly affect the microbial dysbiosis is not solid. please rephrase

ANS: We thank the reviewer for the comment as cut the dysbiosis from the sentence as following (at discussion topic 3.2) “Additionally, S. cerevisiae might manipulate macrophage activation by the be-ta-glucan component on the cell wall.”.

Figure 3:

a)Same comment regarding the statistics. Please add the exact value, method used in the legend.

ANS: We thank the reviewer for the comment and added it accordingly. The exact p values were added in the supplement fig 1.

  1. WGP produced significant amount of IL1-B than yeast extract, while yeast produced more IL-6, any verification for this point?

ANS: We thank the reviewer for the comment. The values (I-1b and IL-6) between yeast extract and WGP were not significantly different. However, this is an interesting observation. Then, we mention this possibility in the new discussion as following “Although glucan might be the main component of the yeast extract, a tendency of the difference in serum cytokines after the administration by yeast extract versus the commercially available WGP (higher serum IL-1β with lower IL-6 after WGP) implied the possible contamination in the in-house yeast extract procedure.”.

Figure 4 and 5

  1. why the authors did not do similar experiments with yeast extract and compare between WGP and yeast extract in vitro?

ANS: We thank the reviewer for the comment. Because the in-house extraction might be not as pure as the commercially available glucan, then we do these experiments only by WGPs. Hence, we mentioned this point in the new discussion as following ‘’ Notably, the tests of macrophage stimulation and cell energy manipulation of the in-house yeast extract were not performed here due to the awareness of the standard of the preparation procedures. However, yeast extract might be an economical source of glucan for the real clinical setting in some developing countries.”.

Round 2

Reviewer 2 Report

No further comments

Author Response

Specific comments are as follows:
1. Abbreviations: The use of abbreviations when writing a manuscript has many advantages
besides simplicity of expression. To use an abbreviation, first write the abbreviation in
parentheses after the full name, and then use the abbreviation from Introduction to the
final conclusion. Only in abstract and figure legend do it separately. Define LPS in the
abstract. Since abbreviation for LPS is not additionally used in the abstract, so it is not
necessary.

ANS: We thank the reviewer for the comment and add lipopolysaccharide before LPS in the abstract.

2. Abstract: cyclic GMP-AMP Synthase should be written as cyclic GMP-AMP synthase.

ANS: We thank the reviewer for the comment and change cyclic GMP-AMP Synthase to cyclic GMP-AMP synthase.

3. References 1 and 2 in Introduction: It is questionable what the statistical data on the
mortality rate from colorectal cancer in 2035 will mean. Rather, the most recent cancer
statistics for 2020 or 2021 are judged to be meaningful. Or, compare the recent number of
colorectal cancer deaths with the number of colorectal cancer deaths in 2035.

ANS: We thank the reviewer for the comment and cut this part of the introduction into “Colorectal cancer is one of the leading causes of cancer-related death worldwide [1, 2].”

4. Page 2, 2.1. in Results: Define WT. Considering the characteristic of IJMS, which places
the Materials and Methods section at the end of the article, it is appropriate to define the
abbreviation used for the first time.

ANS: We thank the reviewer for the comment and add the full name of WT; wildtype in result 2.1.

5. Figure legends for Figure 1 and Figure 2: Azoxymethane should be written as
azoxymethane.

ANS: We thank the reviewer for the comment and change Azoxymethane to azoxymethane in the figure legends for figure1 and 2.

6. Results 2.2.: Define LPS here.

ANS: We thank the reviewer for the comment and add lipopolysaccharide before LPS in the results 2.2.

7. At the end of Figure 3 legend: table 1 should be written as Table 1.

ANS: We thank the reviewer for the comment and change table 1 to Table 1 in the end of figure 3 legend.

8. Saccharomyces cerevisiae at the top of Page 13: Although Saccharomyces cerevisiae is
referred to as S. cerevisiae in the first part of the Introduction, it is referred to as
Saccharomyces cerevisiae here and there. Abbreviated as promised.

ANS: We thank the reviewer for the comment and cut the full name into S. cerevisiae.

  1. Since the unit of temperature is expressed in several ways, unify it as one. Examples:
    2 °C, 35°c, and 37°C.

ANS: We thank the reviewer for the comment and change the pattern of temperature into 2 °C.

10. Materials and Methods section - When naming a particular chemical company, you must
provide location information such as company name, city and/or state (abbreviation in the
USA and Canada) and country. Once you have named a company with the information,
you should only mention a company’s name thereafter. Examples: MoBio, Carlsbad,
California should be written as MoBio, Carlsbad, CA; National Institutes of Health, etc.

ANS: We thank the reviewer for the comment and change the pattern of chemical company.

11. Reference section: Author should consult and peruse carefully recent issues of the
journal, International Journal of Molecular Sciences, for format and style. The first letter
of the title must be in upper case, and the rest must be in lower case. Examples: 3, 7, 8, 10,
12, 15, 16, 19, 23, 24, 28, 31, 32, 34, 36, 38, 40, 41, 42, 44, 47, 49, 50, 52, 53, 54, 56, 62,
64, 65, 68, 70, 72, 73, 78, 81,82, 83, etc

ANS: We thank the reviewer for the comment and change the format and style in reference.